# The Stability of Mean Wood Specific Gravity across Stand Age in US Forests Despite Species Turnover

**Sean P. Healey * and James Menlove**

US Forest Service Rocky Mountain Research Station, Ogden, UT 84401, USA; jmenlove@fs.fed.us
* Correspondence: seanhealey@fs.fed.us; Tel.: +1-801-391-7536

**Abstract:** Research Highlights: Estimates using measurements from a sample of approximately 132,000 field plots imply that while the species composition of US forests varies substantially across different age groups, the specific gravity of wood in those forests does not. This suggests that models using increasingly accurate spaceborne measurements of tree size to model forest biomass do not need to consider stand age as a covariate, greatly reducing model complexity and calibration data requirements. Background and Objectives: Upcoming lidar and radar platforms will give us unprecedented information about how big the trees around the world are. To estimate biomass from these measurements, one must know if tall trees in young stands have the same biomass density as trees of equal size in older stands. Conventional succession theory suggests that fast-growing pioneers often have lower wood (and biomass) density than the species that eventually dominate older stands. Materials and Methods: We used a nationally consistent database of field measurements to analyze patterns of both wood specific gravity (WSG) across age groups in the United States and changes of species composition that would explain any shifts in WSG. Results: Shifts in species composition were observed across 12 different ecological divisions within the US, reflecting both successional processes and management history impacts. However, steady increases in WSG with age were not observed, and WSG differences were much larger across ecosystems than across within-ecosystem age groups. Conclusions: With no strong evidence that age is important in specifying how much biomass to ascribe to trees of a particular size, field data collection can focus on acquiring reference data in poorly sampled ecosystems instead of expanding existing samples to include a range of ages for each level of canopy height.

**Keywords:** biomass; wood density; lidar; forest inventory; stand dynamics

## 1. Introduction

Forest biomass is a globally important carbon reservoir, and forest growth can significantly mitigate the climate-altering effects of fossil fuel emissions [1,2]. There are a number of incipient spaceborne missions designed to measure forest biomass around the planet:

(1) GEDI (NASA's Global Ecosystem Dynamics Investigation) uses waveform lidar to measure aboveground biomass (AGB) at 22-m footprints, which is in turn used to infer mean biomass at the level of 1-km grid cells [3–5];

(2) BIOMASS (a European Space Agency Earth Explorer mission) uses P-band radar to map AGB at 200m spatial resolution [6];

(3) NISAR (joint mission between NASA and the Indian Space Research Organisation) operates an L-band and an S-band synthetic aperture radar that enables observation of biomass change at hectare scales [7]; and,

(4)    ICESAT-2 (NASA) uses linear tracks of pulse-counting lidar that, in combination with other instruments, allows measurement of vegetation biomass [8].

While sensors such as these provide observations closely correlated with forest structure, biomass must be inferred through the use of models calibrated with ground measurements [9–11]. Such models imply, sometimes explicitly [12], a stand-level wood density: a ratio of measured canopy volume to biomass. What these models do not consider, to our knowledge, is any sensitivity of wood specific gravity (WSG) to stand age.

This is surprising for two reasons. First, within individual ecosystems, it is well understood that faster-growing pioneer species often have lower WSG than later-successional trees [13,14]. If young stands are indeed dominated by lower-density species, models applied to remote sensing data from those stands will generally over-predict biomass. This is important because a promising application of globally collected biomass data is inference of ecosystem carbon dynamics by estimating the difference in average carbon content of stands with different ages. This process, sometimes called "chronosequencing" when used with inventory plots [15], provides insight into rates of carbon accumulation.

Over-prediction of the WSG of young stands in this framework would obscure both the impact of stand-replacing disturbances upon carbon stocks and the rate of carbon re-growth. The fact that biomass models are generally not sensitive to stand age is also surprising because long-term, spatially exhaustive records of forest disturbance are widely available [16,17]. Such records could be used to both develop and apply age-sensitive models.

In this paper, we used the nationally comprehensive inventory of the US Forest Service's FIA (Forest Inventory and Analysis) Program to evaluate the degree to which stand-level mean WSG depends upon stand age. FIA maintains a grid of inventory plots (approximately 1 plot per 2430 hectares), across which a variety of tree- and stand-level attributes are measured every 5–10 years. This sample is the basis for the forest component of the US Greenhouse Gas Inventory [18] and can also support detailed assessment of model scaling properties across a wide range of conditions (e.g., [19]). We made estimates of mean basal area-weighted WSG (with uncertainty) for each 10-year age bin within 12 ecological divisions covering the conterminous United States.

To understand any changes in WSG as a function of stand age, we used the same database to evaluate how the proportional distribution (by share of total basal area) of the eight most common species (again, by basal area) changed across age bins in each ecoregion. While there is no reason to see ecosystems of the United States as globally representative, analysis across 12 diverse, data-rich regions was intended to suggest the degree to which stand age's impact on WSG should or should not be considered in efforts to model AGB from remotely sensed measurements.

## 2. Methods

The goal of this study was to estimate, for each of the ecological divisions shown in Figure 1, the mean WSG for forests in ten-year age classes. This required knowing stand age and average basal area-weighted WSG at the level of the FIA condition. Conditions, which are mapped sub-divisions of FIA plots defined by boundaries related to variables such as land cover class and forest type, are the finest unit at which stand age is assigned. Inclusion probability of all conditions within FIA's spatially balanced simple random sample frame are known, so once mean WSG was known for each condition, FIA's standard estimation protocols [20] could be used to derive ecozone-level estimates of WSG by age class.

In this investigation, it was important to include seedlings along with more mature trees in the basal area weighting process. Though seedlings (diameter less than 2.54 cm) have no recorded basal area in the database, such trees can be the only cohort in the critical first age bin (1–10 years old). FIA only measures these trees on micro-plots (2.07 m radius) nested within each of the four sub-plots (7.32 m radius) making up the standard plot design. We calculated basal area for each measured seedling based upon an arbitrarily chosen diameter of 1 cm. We likewise included saplings (2.54–12.7 cm diameter), which are also measured on micro-plots, but do have recorded diameters.

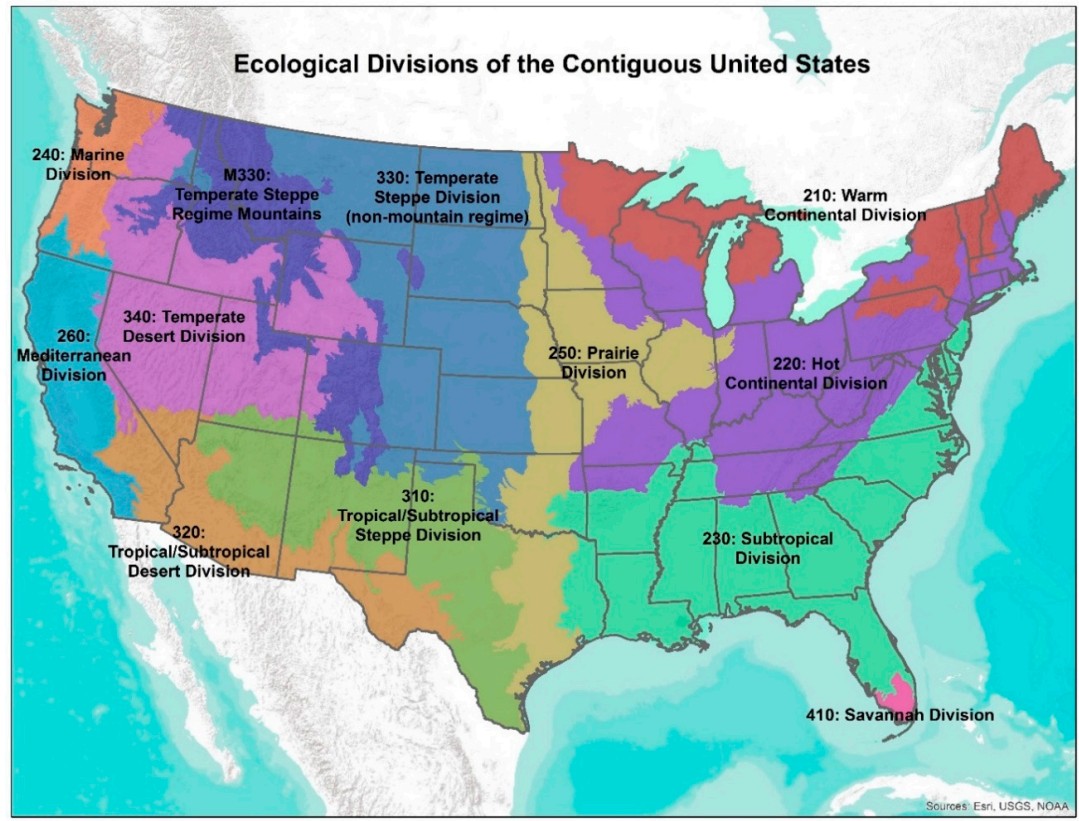

**Figure 1.** Ecological divisions [21] used here to analyze trends in WSG and species turnover across the United States.

Because the probabilities of inclusion for seedlings and saplings observed on micro-plots and trees observed on sub-plots were different, it was necessary to convert (from [22]) the sums of both basal area and the product of basal area and WSG to per-unit-area terms to determine mean basal area-weighted wood density at the condition level:

$$y_{ik} = \frac{\sum_j^4 \sum_t y_{ijkt}}{\sum_j^4 a_{ijk}} + \frac{\sum_j^4 \sum_t y'_{ijkt}}{\sum_j^4 a'_{ijk}} \tag{1}$$

where $y_{ijkt}$ = the attribute of interest associated with tree $t$ on subplot $j$ covering condition $k$ on plot $i$. $a_{ijk}$ = the area used to observe the attribute of interest on subplot $j$ covering condition $k$ on plot $i$. $y'_{ijkt}$ = the attribute of interest associated with tree $t$ on subplot or microplot $j$ covering condition $k$ on plot $i$. $a'_{ijk}$ = the area used to observe the attribute of interest (microplot $j$) covering condition $k$ on plot $i$ and $y_{ik}$ is calculated first where $y$ = basal area ($B$) and then where $y$ = basal area * wood specific gravity ($BG$).

Wood density, $G$, is assessed by FIA at the species level, using values from historical studies [23]. Condition-level basal area-weighted density, $G_{ik}$, is then isolated through:

$$G_{ik} = \frac{(BG)_{ik}}{B_{ik}} \tag{2}$$

Stand age at the condition level is assigned by field crews through coring representative overstory trees at breast height and adding a fixed amount of time to represent the period between stand establishment and pith development at breast height [24]. Mean WSG per 10-year age class was then estimated at the ecological division scale, accounting for the proportion of the area sampled by each condition, following [21]. To provide context around any differences across ages in mean WSG, total basal area by tree species for each age bin was also calculated (ibid.).

## 3. Results

Measurements in this study were based upon 3,548,265 tree measurements and 2,695,735 seedling measurements collected across 132,582 FIA plots (Table 1). The area of each ecological division ranged from 0.4 to 67.9 million hectares (ha). The overall basal-weighted mean WSG varied by region from 0.4 (Mountain Regime of the Temperate Steppe division) to 0.65 (Tropical/Subtropical Desert).

**Table 1.** Sample properties and results by Ecological Division.

| Division | FIA Estimate of Forestland (million ha) | Measured Trees/Seedlings | Measured Plots | Number of Surveyed Species | Mean Specific Gravity Estimate |
|---|---|---|---|---|---|
| Warm Continental (210) | 35.6 | 733,108/821,979 | 21,196 | 177 | 0.45 |
| Hot Continental (220) | 47.8 | 617,470/571,091 | 25,031 | 194 | 0.52 |
| Subtropical (230) | 67.9 | 919,398/551,384 | 32,217 | 204 | 0.50 |
| Marine (240) | 13.8 | 311,115/149,696 | 9371 | 63 | 0.42 |
| Prairie (250) | 10.9 | 81,551/72,208 | 4883 | 156 | 0.56 |
| Mediterranean (260) | 14.2 | 164,136/65,526 | 5315 | 78 | 0.44 |
| Tropical/Subtropical Steppe (310) | 27.9 | 153,967/77,679 | 9888 | 188 | 0.59 |
| Tropical/Subtropical Desert (320) | 6.5 | 18,572/8268 | 2168 | 73 | 0.65 |
| Temperate Steppe (330) without Mountain Regime | 4.5 | 27,468/23,904 | 1823 | 93 | 0.45 |
| Temperate Steppe (330) just Mountain Regime | 35.2 | 417,743/292,708 | 15,411 | 63 | 0.4 |
| Temperate Desert (340) | 12.5 | 99,703/56,268 | 5140 | 41 | 0.54 |
| Savannah (410) | 0.4 | 4034/5025 | 139 | 37 | 0.48 |

Significant shifts in species distributions occurred across age classes in each ecological division. Figure 2 shows these shifts in terms of the changing fraction of total basal area by species across age bins (limited to eight most common species by basal area); scientific names of all species are listed in Appendix A. In some cases, species are well represented either in the younger or older age groups. For example, in the Tropical/Subtropical Desert division, honey mesquite is dominant in the younger age classes but virtually absent in the older classes, while Utah juniper shows the opposite trend. A species' fractional basal area across age bins may also remain relatively stable (e.g., western hemlock in the Marine division), or it may be multi-modal (e.g., loblolly pine in the Subtropical division). The WSG assigned by FIA to the species shown in Figure 2 varies from 0.29 (northern white cedar) to 0.8 (live oak).

Despite the dynamic distribution of tree species by age class, basal area-weighted WSG remained relatively constant in most regions. The only place where WSG trended consistently upward over age groups was the Hot Continental division, and even there the mean difference between the 1–10-year-old age group and the over 171-year-old group was just 0.05. Sampling errors were small relative to estimated mean specific gravity, with the exception of some of the rarer, older age groups in the Tropical/Sub-tropical Desert and the Temperate Steppe (non-mountain) divisions (Figure 2).

The largest change in WSG across age groups at the division level was actually a decline with age; mean density went from 0.69 at 11–20 years to 0.48 at 101–110 years in the Tropical/Subtropical Steppe division. Changes in most ecological divisions were much smaller, and in no case was there a sharp discontinuity in density from younger to older stands. The range of average specific gravity was greater across ecological divisions, ranging from 0.4 to 0.65 (Table 1), than within ecological divisions across age groups (Figure 2).

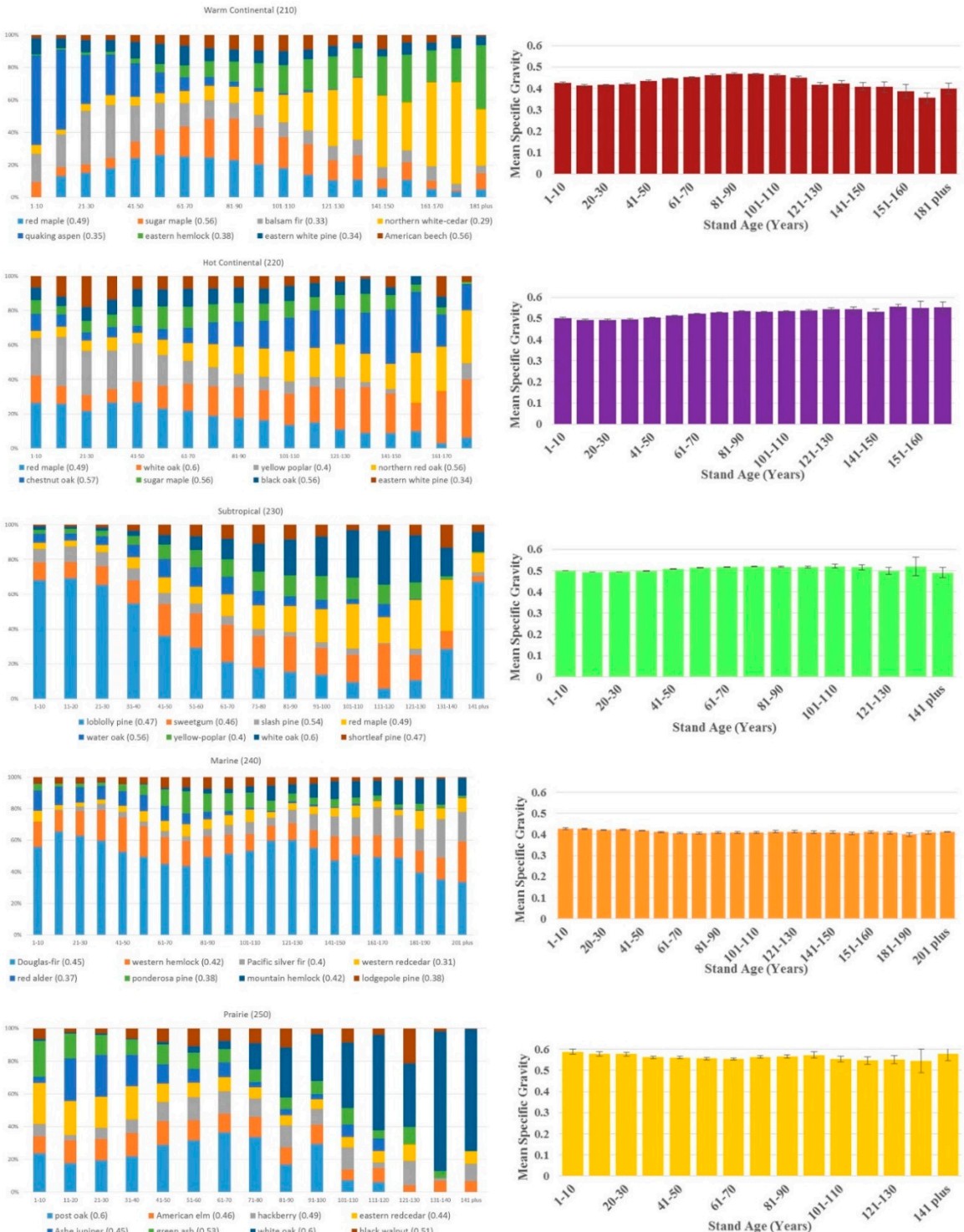

**Figure 2.** *Cont.*

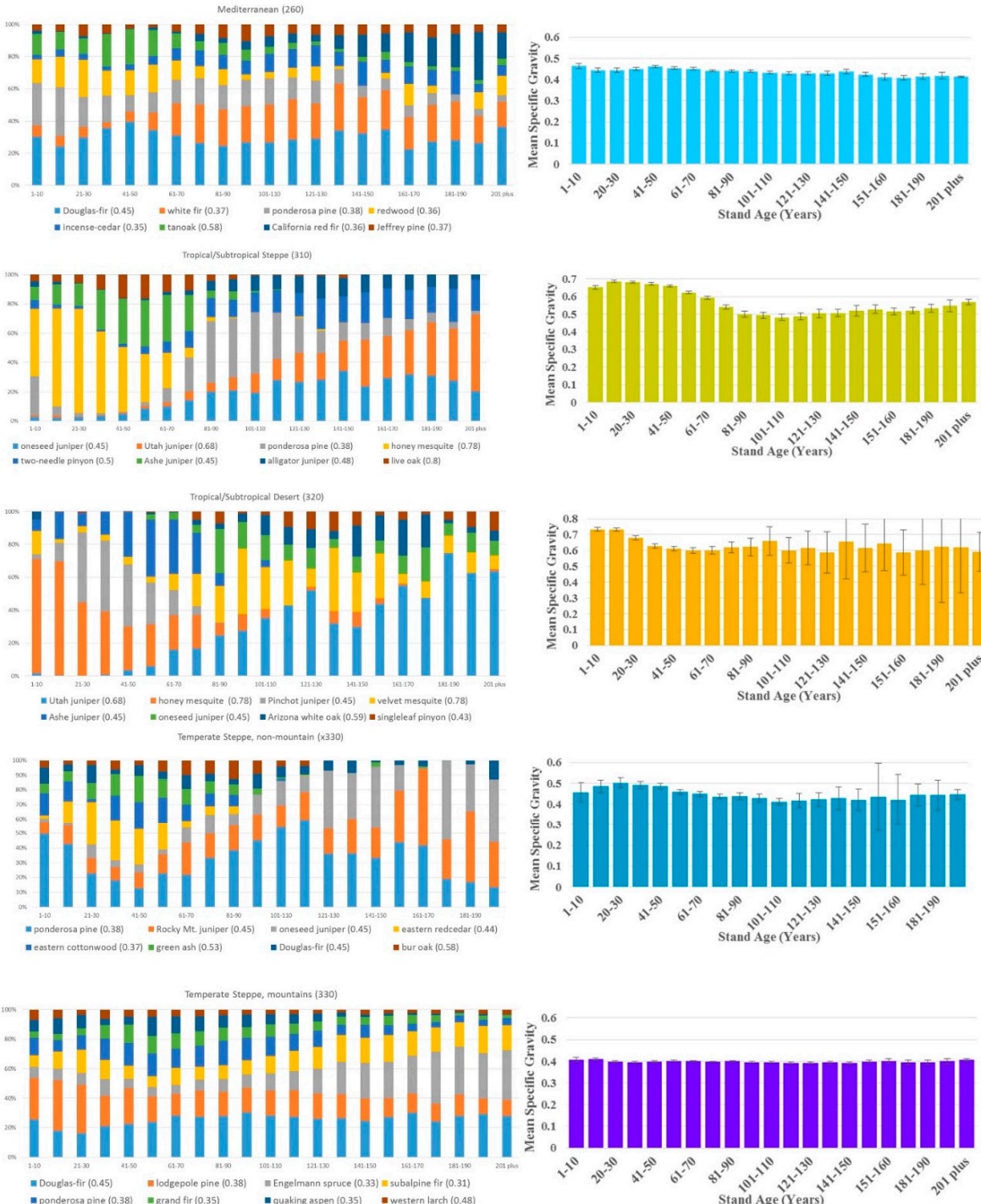

**Figure 2.** *Cont.*

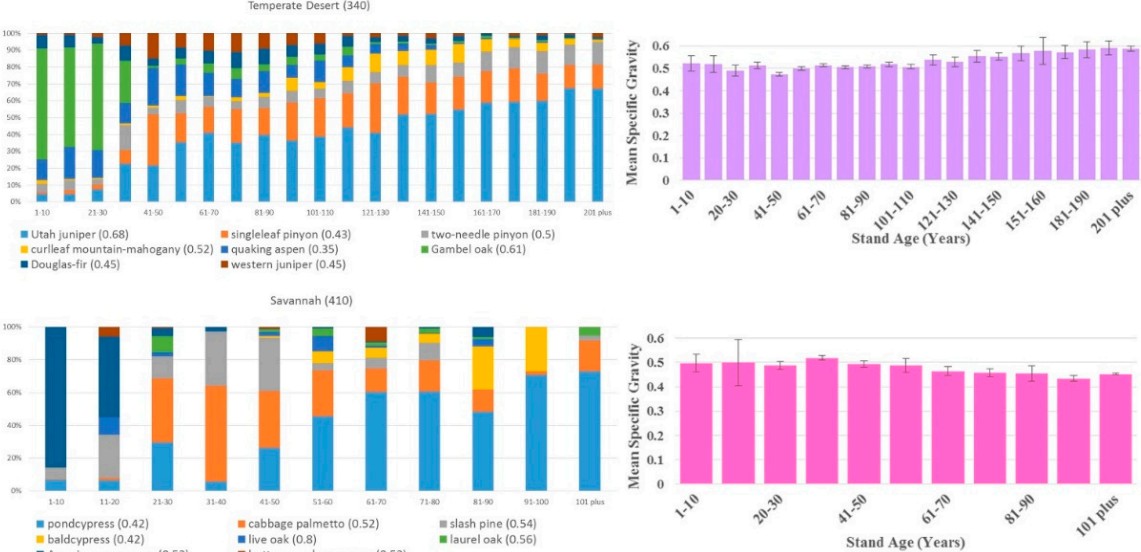

**Figure 2.** Species distribution by basal area across age groups of the eight most common species in each ecological division (**left**) and the division's basal area-weighted mean wood specific gravity across age, considering all species (**right**). Error bars represent FIA sampling error. Bar color on the right matches the division's color on the map in Figure 1.

## 4. Discussion

Understanding the rate of AGB recovery following disturbance is critical for predicting forest carbon dynamics and for identifying the potential impact of forest management in mitigating climate change. Support for such assessments is relatively simple when extensive inventory data are available; designed samples used by inventories promote representativeness, and measurement error of field samples is generally presumed to be negligible. Williams et al. [25] refined net ecosystem productivity (NEP) functions used in a process model (the Carnegie-Ames-Stanford Approach) by plotting measured AGB against stand age at a large number of FIA locations. This allowed improved characterization of the role of disturbance and management in carbon dynamics of US forests.

Additional uncertainties become relevant when basic data about biomass and biomass change must be modeled from remote sensing. For example, vegetation demographic models, which track multiple size classes within a given plant functional type to predict outcomes under different scenarios, can use remotely sensed forest biomass maps as benchmark constraints [26]. One type of uncertainty involved with using remote sensing as a benchmark involves transferability: does the relationship used to build the underlying model—between biomass and lidar- or radar-measured height, for example—actually hold in the populations where the model is applied? There is broad awareness of this issue in the context of transferring models across regions [27]. The issue is less appreciated when models are built locally but might not be appropriate for transfer across different successional stages. This is particularly relevant if the difference in predicted biomass between old and young stands is being used to infer a rate of biomass accumulation.

The question addressed by this paper was relatively narrow: with an emerging generation of active remote sensing platforms capable of supporting ecosystem demography and other approaches to tracking carbon dynamics, should models targeting prediction of forest biomass explicitly account for stand age? Remote sensing generally only measures canopy structure, requiring models to either implicitly or explicitly assign the wood density (measured by specific gravity) needed to infer biomass. If WSG were found to change radically with stand age, perhaps due to successional processes, it would argue for broad use of age-sensitive models.

Given the relatively high precision of our estimates (approximately 500 field plots per age bin per region), it is clear that shifts in WSG do occur with stand age (Figure 2). However, our results

suggested that differences in WSG at the ecological division scale are much lower across stand age than they are across ecosystems. With field resources scarce in many regions, this argues for prioritizing collection of new data in under-sampled ecosystems above sampling a range of ages for each tree size category.

Our analysis treated WSG as a species-specific constant, ignoring variation that might occur due to soil properties [28] or stand density [29]. However, site-level variation probably had a minimal effect across such a large sample. More serious with respect to our goal of comparing wood density across age groups would be a case where the specific gravity of wood added by individual trees changes as the trees age. Investigations into this question, though, have shown wood density to vary only slightly as a function of tree age and factors such as competition that may co-vary with tree age [30,31]. Treating WSG as a constant also ignores variation that may occur across ecological gradients. There is evidence that, both within and across species, WSG can vary with climatic factors such as temperature and precipitation [32,33]. Assigning more regionally specific density values (assuming they are correct) for species such as Douglas fir, which range from moist lowland forests to drier high-elevation systems, would lead to more accurate estimates of mean WSG across age groups. Until nationally comprehensive empirical data exist that can be used to refine the WSG values used by FIA, the impact of local ecological variation will remain unknown.

It is reasonable to ask why it appears mean WSG does not increase with age, as one might expect from an idealized scenario where fast-growing colonizer species are replaced by slower-growing, longer-lived trees. This dynamic undoubtedly occurs on some sites, and species composition does vary strongly by age in many of the regions studied. In some cases, succession simply does not produce a change in average WSG. For instance, red alder, which fixes nitrogen and is an early colonizer following disturbance [34], occurred in younger stands in the Marine division, but not in older stands. At the same time, slower-growing, shade tolerant species such as mountain hemlock and silver fir represented a significant share of basal area only in the region's older stands. However, the wood of red alder, silver fir, and mountain hemlock all have specific gravities between 0.37 and 0.42; succession in this region did not bring a big change in mean wood density.

More broadly, the role of human impact must be acknowledged. Age bins in Figure 2 do not represent uninterrupted successional pathways; management of regeneration and historical harvest patterns have manipulated species distributions for centuries. For example, loblolly pines were relatively rare in the pre-European forests of the southern United States. The collapse of the cotton industry in the 1880s provided abandoned fields where loblolly pine had an establishment advantage because of their light, easily dispersed seeds [35]. Fire suppression allowed these new loblolly stands to thrive in the following decades [ibid], explaining the species' importance in the region's older age classes. Loblolly pine is also widely used in plantations, where it is managed on relatively short rotations. This complex history explains the species' multi-modal age distribution in Figure 2, and more importantly illustrates how human intervention affects the demography of our forests.

This study notably did not cover a tropical system, and work in other areas should assess whether our results—the relative insensitivity of WSG at regional scales to stand age—apply elsewhere. While tropical species such as cecropia fit the stereotype of low-density colonizers [36], many disturbed tropical sites are colonized by longer-lived pioneers with life forms closer to those of primary forest trees [37]. Human intervention may likewise disrupt presumed successional trends, as was the case with loblolly pine in the Subtropical Division.

## 5. Conclusions

In general, it is convenient that average WSG—the amount of biomass per unit volume—does not vary strongly with stand age. New remote sensing platforms are giving us unprecedented information about tree size, but we would need significantly more calibration data, and more complex models, if large trees in young stands had consistently higher or lower biomass than large trees in older stands. Differences in WSG were much greater across ecosystems than across age groups within the

same ecosystem. This finding allows the community to focus upon acquiring reference data in poorly sampled ecosystems instead of expanding existing samples to include a range of ages for each level of canopy height.

**Author Contributions:** The study was conceived by S.P.H. and designed by J.M. J.M. was responsible for all FIA analysis, while the manuscript was written mostly by S.P.H.

**Funding:** This research was funded by NASA's Carbon Monitoring System: grant 80HQTR18T0016 (solicitation NNH16ZDA001N). Further support (Healey's salary) was provided by the US Forest Service Rocky Mountain Research Station Inventory and Monitoring Program.

**Conflicts of Interest:** The authors declare no conflict of interest.

## Appendix A. Scientific Names for Cited Tree Species

| Common Name | Latin binomial | Common Name | Latin binomial |
|---|---|---|---|
| Pacific silver fir | *Abies amabilis* (Dougl. ex Louden) | lodgepole pine | *Pinus contorta* Douglas ex Loud.) |
| balsam fir | *Abies balsamea* (L.) Mill. | two-needle pinyon | *Pinus edulis* Engelm. |
| white fir | *Abies concolor* (Gord. & Glend.) Lindl. ex Hildebr. | slash pine | *Pinus elliottii* Engelm. |
| grand fir | *Abies grandis* (Dougl. ex D. Don.) Lindl. | Jeffrey pine | *Pinus jeffreyi* Balf. |
| subalpine fir | *Abies lasiocarpa* (Hook.) Nutt. | singleleaf pinyon | *Pinus monophylla* Torr. & Frém. |
| California red fir | *Abies magnifica* A. Murr. | ponderosa pine | *Pinus ponderosa* C. Lawson |
| bigleaf maple | *Acer macrophyllum* Pursh | eastern white pine | *Pinus strobus* L. |
| red maple | *Acer rubrum* L. | loblolly pine | *Pinus taeda* L. |
| sugar maple | *Acer saccharum* L. | eastern cottonwood | *Populus deltoids* Bartram ex Marsh. |
| red alder | *Alunus rubra* Bong. | quaking aspen | *Populus tremuloides* Michx. |
| yellow birch | *Betula alleghaniensis* Britton | honey mesquite | *Prosopis glandulosa* Torr. |
| incense cedar | *Calocedrus decurrens* (Torr.) Florin | velvet mesquite | *Prosopis velutina* Woot. |
| hackberry | *Celtis occidentalis* L. | black cherry | *Prunus serotine* Ehrh. |
| mountain-mahogany | *Cercocarpus ledifolius* Nutt. | Douglas-fir | *Pseudotsuga menziesii* |
| buttonwood-mangrove | *Conocarpus erectus* L. | white oak | *Quercus alba* L. |
| American beech | *Fagus grandifolia* Ehrh. | Arizona white oak | *Quercus arizonica* Sarg. |
| white ash | *Fraxinus americana* L. | canyon live oak | *Quercus chrysolepis* Liebm. |
| green ash | *Fraxinus pennsylvanica* Marsh. | southern red oak | *Quercus falcata* Michx. |
| black walnut | *Juglans nigra* L. | Gambel oak | *Quercus gambelii* Nutt. |
| Ashe juniper | *Juniperus ashei* J. Buchholz | laurel oak | *Quercus laurifolia* Michx. |
| alligator juniper | *Juniperus deppeana* Steud. | bur oak | *Quercus macrocarpa* Michx. |
| redberry juniper | *Juniperus coahuilensis* (Martiñez) Gausen ex R.P. Adams | chestnut oak | *Quercus montana* Willd. |
| oneseed juniper | *Juniperus monosperma* (Engelm.) Sarg. | water oak | *Quercus nigra* L. |
| western juniper | *Juniperus occidentalis* Hook. | northern red oak | *Quercus rubra* L. |
| Utah juniper | *Juniperus osteosperma* (Torr.) Little | post oak | *Quercus stellate* Wangenh. |
| Pinchot juniper | *Juniperus pinchotii* Sudworgh | black oak | *Quercus velutina* Lam. |
| Rocky Mountain juniper | *Juniperus scopulorum* Sarg. | live oak | *Quercus virginiana* Mill. |
| eastern redcedar | *Juniperus virginiana* L. | American mangrove | *Rhizophora mangle* L. |
| western larch | *Larix occidentalis* Nutt. | cabbage palmetto | *Sabal palmetto* (Walter) Lodd. ex Schult. & Schult. f. |
| sweetgum | *Liquidambar styraciflua* L. | redwood | *Sequoia sempervirens* (Lamb. ex D. Don.) Endl. |
| yellow poplar | *Liriodendron tulipifera* L. | pond cypress | *Taxodium ascendens* Brongn. |
| tanoak | *Lithocarpus densiflorus* (Hook. & Arn.) Rehd. | baldcypress | *Taxodium distichum* (L.) Rich. |
| melaleuca | *Melaleuca quinquenervia* (Cav.) S.F. Blake | northern white-cedar | *Thuja occidentalis* L. |
| swamp tupelo | *Nyssa biflora* Walter | western redcedar | *Thuja plicata* ) Donn ex D. Don |
| redbay | *Persea borbonia* (L.) Spreng. | eastern hemlock | *Tsuga Canadensis* L. |
| Engelmann spruce | *Picea engelmannii* Parry ex Engelm. | western hemlock | *Tsuga heterophylla* (Raf.) Sarg. |
| red spruce | *Picea rubens* Sarg. | mountain hemlock | *Tsuga mertensiana* (Bong.) Carrière |
| shortleaf pine | *Pinus echinata* Mill. | American elm | *Ulmus Americana* L. |

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
