# Peer review of "The Stability of Mean Wood Specific Gravity across Stand Age in US Forests Despite Species Turnover"

_forests, doi:10.3390/f10020114_

Round 1
Reviewer 1 Report
I have studied the paper with much attention and I confirm that paper has merit but needs a major revision and so I suggest to reject and to invite the authors to submit again.
Abstract need a revision to improve the comprehension of the work. I'll suggest to remove the section’ title (M/M-Results) and reduce the results and explain the contest and the discussion.
I'm not sure whether the content of the article is sufficient for publishing in this journal with such a high impact factor. In my opinion, the experiment is poor and need to be spreaded. However, it is definitely necessary to supplement it with additional information for publishing.
In scientific papers, the third person is favored.
Introduction section is too short;
Methodology is not correctly explain the method and is very short
The results are not explain very well and are very short
The discussion repeat the results, not well focused on your findings and know is possible to develop this new approach.
Some parts of the discussion seems more political than technical and there is not a Conclusion’ paragraph.
Author Response
Please see the attached PDF, explaining revisions based upon comments of each reviewer. Many thanks for the constructive suggestions.

Reviewer 2 Report
Abstract:
Line 10/11: Wood density was not measured. Please rephrase.
Line 16: "Conventional succession theory suggests that fast-growing pioneers often have lower wood (and biomass) density than trees of the same size in older stands." Does this refer to the change of species composition with the age of the stand or to the wood density of the same pioneer species with the age of a tree? (Conventional succession theory suggests that pioneer species dominate young stands, but silvicultural treatments tend to favor more valuable tree species shifting the species composition with the age of the stand.)
Methods and Results:
Referring to different units for expressing wood density in the manuscript is quite confusing (for example: wood density – biomass density – specific density – specific gravity; stand-level wood density as a ratio of measured canopy volume to biomass). I suggest that nominal wood density (oven-dry mass/volume of green wood) - that is commonly used for calculating volume to biomass - or specific gravity (as in the results) should be used consistently throughout the manuscript.
Discussion:
Variation of the wood density of the same wood species in different ecological conditions (and it's possible influence on the final results) should be elaborated (with citations) in more detail (as it is not covered by the study design).
Author Response
Please see the attached pdf for revisions made in response to each reviewer comment. Many thanks for the constructive suggestions.

Reviewer 3 Report
The article is based on the national inventory of the US Forest Service’s FIA Program. Authors analyse the mean wood density for forests across the whole country.
Detailed comments:
1. The title should be shortened to “Average specific gravity of wood in US forests” because the present one is rather a statement.
2. The affiliation of authors should be given only once. They work in the same institution.
3. There is no need to give subtitles like Research Highlights, Background and Objectives etc. in the “Introduction”.
4. In my opinion the “Introduction” is too short and should give more information about background of the problem.
5. Figure 1 should be moved to the part connected with methodology.
6. The resolution of Figure 2 is too low.
7. Conclusions should be separated.
8. In “References” – line 251 - there is a need to give number of volume and pages (no x-xx). It is Remote Sensing 10(11):1832.
Author Response

(The authors gave the same response as above.)

Round 2
Reviewer 1 Report
The paper had been significantly improved by the authors - congratulations to you.
This remains an important research topic and I appreciate the work of the authors.
The authors did improve the suggested review significantly.
Reviewer 3 Report
The article can be published in present form.